# Validation and Classification of the 9-Item Voice Handicap Index (VHI-9i)

**DOI:** 10.3390/jcm10153325

**Published:** 2021-07-28

**Authors:** Felix Caffier, Tadeus Nawka, Konrad Neumann, Matthias Seipelt, Philipp P. Caffier

**Affiliations:** 1Department of Audiology and Phoniatrics, Charité-Universitätsmedizin Berlin, Charitéplatz 1, D-10117 Berlin, Germany; felix.caffier@charite.de (F.C.); tadeus.nawka@charite.de (T.N.); 2Institute of Biometry and Clinical Epidemiology, Charité-Universitätsmedizin Berlin, Campus Charité Mitte, Charitéplatz 1, D-10117 Berlin, Germany; konrad.neumann@charite.de; 3Department of Otorhinolaryngology, Ernst von Bergmann Klinikum Potsdam, Charlottenstr. 72, D-14467 Potsdam, Germany; MatthiasSeipelt@web.de

**Keywords:** Voice Handicap Index (VHI-9i), international short scale, VHI-9i severity levels, test–retest reliability, validation of classification ranges, self-assessed vocal impairment (VHIs), hoarseness, dysphonia severity categories, voice diagnostics

## Abstract

The international nine-item Voice Handicap Index (VHI-9i) is a clinically established short-scale version of the original VHI, quantifying the patients’ self-assessed vocal handicap. However, the current vocal impairment classification is based on percentiles. The main goals of this study were to establish test–retest reliability and a sound statistical basis for VHI-9i severity levels. Between 2009 and 2021, 17,660 consecutive cases were documented. A total of 416 test–retest pairs and 3661 unique cases with complete multidimensional voice diagnostics were statistically analyzed. Classification candidates were the overall self-assessed vocal impairment (VHIs) on a four-point Likert scale, the dysphonia severity index (DSI), the vocal extent measure (VEM), and the auditory–perceptual evaluation (GRB scale). The test–retest correlation of VHI-9i total scores was very high (r = 0.919, *p* < 0.01). Reliability was excellent regardless of gender or professional voice use, with negligible dependency on age. The VHIs correlated best with the VHI-9i, whereas statistical calculations proved that DSI, VEM, and GRB are unsuitable classification criteria. Based on ROC analysis, we suggest modifying the former VHI-9i severity categories as follows: 0 (healthy): 0 ≤ 7; 1 (mild): 8 ≤ 16; 2 (moderate): 17 ≤ 26; and 3 (severe): 27 ≤ 36.

## 1. Introduction 

A patient’s self-assessment of his or her own voice is an important tool for diagnosing voice disorders and vocal treatment outcomes [1,2]. Only the patients themselves can quantify how much a voice disorder impacts their daily lives. For instance, mild hoarseness affects professional voice users such as opera singers in a different way than non-professional voice users such as office workers [3,4]. 

The Voice Handicap Index (VHI) was developed and validated as a statistically robust method to measure the subjective impact of voice disorders [5]. The original questionnaire consists of 30 items (VHI-30) addressing functional, physical and emotional impairments in the context of dysphonia according to the patient’s own experience. Each question is answered on a scale from 0 (never) to 4 (always), resulting in an overall score ranging from 0 to 120. The VHI-30 was translated and validated cross-culturally to form international variants (e.g., [6,7,8,9,10,11]) which were proven to be equivalent with each other [12,13]. 

From our own clinical experience, many patients and medical staff perceive the original 30-item questionnaire as rather time-consuming. To increase overall acceptance and practicability, shortened versions with fewer items were developed. A 12-item questionnaire [14,15] was soon followed by another reduction to 10 items [16,17]. Since 2009, the commonly used variant at the Charité-Universitätsmedizin Berlin is the VHI-9i international questionnaire [14]. It consists of only nine items, after item reduction based on the original VHI-30 and European translations. A detailed discussion of the item and scale development can be found in the original VHI-9i publication [14]. In everyday diagnostic practice, the German translation of the VHI-9i is widely used by laryngologists and phoniatricians in German-speaking countries (e.g., [18,19,20,21,22]). Despite its clinical adoption, the reliability and validity of this VHI short scale as well as its classification have not yet been statistically verified. Instead, the current classification scale is based on the 25th, 50th, and 75th percentiles, dividing the scores into four severity classes. Thus far, clinical experience seems to plausibly reflect the self-perceived voice impairment. However, to overcome this arbitrary percentile-based exploration, we looked for a sound statistical basis for VHI-9i severity levels by revising the current cut-off points. In the context of expert opinion, thorough classifications of vocal parameters are essential for the assessment of dysphonia. In addition, a reliable and valid VHI-9i severity classification is needed to improve clinician-rated evaluations of treatment outcomes (e.g., better characterization of the quantified extent of subjective vocal impairment, more comprehensible assessment of individual pre- vs. post-therapeutic comparisons). 

This study aims to address these shortcomings. Initially, we investigated whether the VHI-9i produces reliable results independent of age, gender or professional voice use. Next, the questionnaire validity was examined. For this purpose, the relationship between VHI-9i total scores and other established vocal parameters was statistically analyzed to establish cut-off values for healthy voices and mild to severe dysphonia. For external criteria, we intended to use objective acoustic–aerodynamic voice function diagnostics including voice range profile (VRP) measurements, dysphonia severity index (DSI) and vocal extent measure (VEM) calculations, as well as the subjective auditory–perceptual evaluation of voices by experienced examiners (GRB scale). Furthermore, the overall self-assessed vocal impairment (VHIs) served as an internal criterion. 

## 2. Materials and Methods 

### 2.1. Study Design and Patients 

This study was conducted in accordance with the Declaration of Helsinki and approved by the local ethical review board. Selection criteria involved informed consent and the completion of the standard phoniatric examination procedures. After taking the medical history, all patients presenting in the Department of Audiology and Phoniatrics, Charité-Universitätsmedizin Berlin, Germany, received a digital videolaryngostroboscopy to assess the laryngeal findings and to establish a medical diagnosis. Subsequently, multidimensional voice function diagnostics were carried out as recommended by the European Laryngological Society (ELS) [1], starting with subjective evaluations (GRB, VHI-9i) and followed by objective voice function diagnostics (VRP, DSI, VEM). For subjective vocal self-assessment, patients completed the VHI-9i questionnaire. To estimate the voice use of every study participant, we also asked about their occupation and categorized them according to Koufman and Isaacson [23]: elite vocal performers (Level 1; e.g., actors, singers, voice artists), professional voice users (Level 2; e.g., teachers, politicians, moderators), non-vocal professionals (Level 3; e.g., lawyers, medical personnel, civil service employees), and non-vocal non-professionals (Level 4; e.g., IT staff, office workers, mechanics). 

Between May 2009 and March 2021, a total of 17,660 consecutive cases were documented in the clinical database. To analyze the reliability of the VHI-9i, 718 patients were asked to complete the same questionnaire for a second time, without therapeutical intervention. The retest form had to be returned within one week to study the differences between the original answers and the retest. The second VHI-9i questionnaire was returned by 517 patients, corresponding to a response rate of 72%. Some questionnaires containing unanswered items or ambiguous checkmarks (e.g., between items) had to be excluded, resulting in 416 test–retest pairs. 

The remaining 16,942 consecutive cases were analyzed to establish the validity of the questionnaire and to calculate statistically valid classification ranges. Since the VHI-9i should be compared with other established vocal parameters, only 7766 cases with complete multi-dimensional diagnostic assessment were considered. Cases with unreliable perturbation measures (jitter > 5%) were excluded, as recommended in the literature [1,24], resulting in a sample size of 6882. After another exclusion of follow-up visits, 3661 complete and unique cases were left for statistical analysis. 

### 2.2. Subjective Examination Instruments 

The VHI-9i represents an item-reduced short scale of the established VHI-30 [14], available in several languages (i.e., Dutch, English, French, German, Italian, Portuguese and Swedish). In this study, the German translation of the questionnaire was used (see Appendix A). Study participants were asked to answer all 9 items on a scale from 0 to 4 (0: never, 1: almost never, 2: sometimes, 3: almost always, 4: always), resulting in a total score between 0 and 36. The total score was then assigned to one of four dysphonia severity categories, ranging from 0 (healthy; 0 ≤ 5), 1 (mild; 6 ≤ 13), 2 (moderate; 14 ≤ 22), to 3 (severe; 23 ≤ 36). However, these categories correspond to a classification proposed by Nawka et al., based on the percentiles of a representative investigation of 716 patients [25]. Since these classification ranges have not yet been validated, statistical calculation of potential cut-off values for the VHI-9i classification was a main goal of this study. 

Additionally, participants were asked to rate their overall voice impairment at present on a scale from 0 to 3 (0: normal, 1: mild, 2: moderate, 3: severe), the VHI summary assessment (VHIs). This index allows patients to assess how they feel about their voice with only one number. The relationship between VHI-9i and VHIs scores was examined to determine whether patients would rate themselves differently when asked about specific situations in their lives (VHI-9i items) or directly about their overall impairment (VHIs). 

Apart from self-assessment, voices were also evaluated by auditory–perceptual assessment using the GRB system [26,27,28]. Based on the GRBAS scale, our department developed the modified GRB classification [29,30]. Only the first three criteria are used, focusing on the overall grade of hoarseness (G) and both main pathophysiological hoarseness components: roughness (R) and breathiness (B). The assessment of voice quality can be carried out more quickly and easily. Therefore, this system has become established in German-speaking countries and is also recommended in the ELS protocol [1]. Patients were asked to read the standardized text “The north wind and the sun” (German version), while the perceived G, R and B were scored on a scale from 0 to 3. To increase objectivity, each voice recording was rated independently by one experienced phoniatric physician and one senior speech–language therapist. The means were used for further exploration. While the degree of G serves as the overall indicator of dysphonia in the original GRBAS scale, it is regarded as gold standard for hoarseness evaluation in the GRB system presented here [31]. 

### 2.3. Objective Acoustic Assessment 

For objective external validation criteria, we applied acoustic–aerodynamic voice function diagnostics. Voice recordings of all participants were conducted at the voice lab of our outpatient department, which is a sound-treated room with a background noise <40 dB(A). Study participants were asked to wear a head-mounted microphone with a stable mouth–microphone distance of 30 cm [32]. The equipment used for this purpose was the XION microphone headset (model number 352,009,010; XION GmbH, Berlin, Germany), which enables the realization of speech and singing VRP measurements and voice analyses under reproducible conditions. Technical microphone specifications include a frequency response of 70 Hz–20 kHz and a dynamic range of 40–120 dB(A). The microphone headset incorporates a calibrated audio interface that transmits digitized data to the PC via USB. The built-in electronics ensure the automatic calibration of the microphone connection without additional adjustments. The audio was processed via the DiVAS 2.8 software using the Singing Voice Analysis module (product number 350,020,013) and the Speaking Voice Analysis module (product number 350,020,024; XION GmbH, Berlin, Germany). VRP measurements were performed to show the functional interactions of different components of voice generation regarding vocal frequency and intensity [33,34]. The detailed procedure of VRP recordings is described in previous publications [35,36].

The established parameter DSI was automatically calculated as a weighted combination of the highest possible fundamental frequency, the lowest phonation intensity, maximum phonation time and jitter [37]. Regarding jitter, the waveform matching method was used for fundamental frequency extraction as it meets the high-precision criterion of being able to extract a 1% frequency change per cycle with a 1% accuracy, as long as the signal-to-noise ratio is greater than about 40 dB and concomitant amplitude modulations are below about 5% [24]. Measurements were conducted in a standing position. Subjects were asked to produce a sustained vowel (/na/ or /a/) for about 3 seconds at comfortable pitch and loudness. The most stable recording out of 3 trials was chosen for DSI calculation. Based on Gonnermann’s investigation of 495 subjects [38], the DSI scores were sorted into 4 severity categories, discriminating healthy voices (≥4.2) from mildly (<4.2 to ≥1.8), moderately (<1.8 to ≥−1.2), or severely (<−1.2) dysphonic voices. Since the DSI quantifies dysphonia as a negative criterion and involves the risk of imprecise results due to its multidimensional data acquisition, the one-dimensional parameter VEM was recently developed [35]. 

VEM calculation was performed automatically after VRP recording via the proprietary AVA software [39,40]. The VEM quantifies a subject’s dynamic performance and frequency range. It is calculated as a relation of the area and perimeter of the VRP and describes the vocal function by an interval-scaled value without unit, usually between 0 and 120. These limits may be exceeded at both ends by either severely impaired or exceptionally capable voices with a large ambitus and dynamic range. A small vocal capacity is described by a low VEM, a large VRP by a high VEM. The VEM emphasizes the vocal abilities and enables a classification of voice performance as a positive criterion [21,31,41]. Based on Müller’s investigation of 994 subjects [36], the resulting VEM scores were divided into percentiles, distinguishing a normal vocal capacity (≥108) from mildly reduced (<108 to ≥93), moderately (<93 to ≥69) and severely reduced (<69) vocal capacities. 

Table 1 summarizes the severity classification of different objective and subjective vocal parameters by reference range. In contrast to the ordinally scaled GRB and VHIs, the classifications of metrically scaled parameters (VEM, VHI-30, VHI-9i) are based on the percentiles of the respective study cohorts (Level 0: 100th percentile/4th quartile; Level 1: 75th percentile/3rd quartile; Level 2: 50th percentile/2nd quartile; Level 3: 25th percentile/1st quartile). 

## 3. Data Analysis 

Statistical analysis was performed using IBM SPSS version 26.0.0.1. To establish the questionnaire as reliable, the absolute differences in total VHI-9i scores between test and retest were compared. An analysis of the differences of every single item in the questionnaire is individually important, but only the total scores are relevant in diagnostic practice. Paired-sample *t*-tests were used to check for biases, and correlations were established through Pearson’s r. To test the dependency of the VHI-9i total score on age, a regression analysis was performed. Gender differences were analyzed through independent sample t-tests. We checked for a dependency on voice use by means of the nonparametric Kruskal–Wallis H-test. 

Before the cut-off points for the VHI-9i severity categories could be validated, the correlations between the VHI-9i and the severity classifications for VHIs, DSI, VEM, G, R and B had to be determined using Spearman’s rho (ϱ), in order to choose which of them was best suited for classification. These vocal parameters had to be balanced in terms of sensitivity (i.e., true positive rate, TPR) and specificity (i.e., true negative rate, TNR) when applied to the VHI-9i scores. Receiver operator characteristic (ROC) curves were used, which plot the TPR against the false positive rate (FPR = 1 − TNR). Since ROC is a binary classifier, the curves had to be plotted three times to establish possible cut-off points for every severity level (0 vs. 1–3, 0–1 vs. 2–3, 0–2 vs. 3). The area under the curve (AUC) was used to rank the performance of every curve to distinguish between two severity classes. Values between 0.8 and 0.9 are considered excellent, 0.7 to 0.8 acceptable, 0.5 to 0.7 poor.

Several methods exist to determine good class boundaries from ROC curves. As a starting point, we used Youden’s index (J) [42]. The highest J (Max J) is achieved when sensitivity and specificity are at optimal balance (J = TPR − FPR = TPR + TNR − 1). As a second possible class boundary, we determined the point where the number of correctly classified cases (CCCs) was the highest. The CCC is calculated as follows:



CCC= TPR* (n cases of classifying index above class boundary)+ TNR* (n cases of classifying index below class boundary)


To find plausible cut-off values or categories of reasonable size, we selected a value between the two suggested class boundaries based on the median between Max J and Max CCC, also taking into account well over a decade of clinical experience with the VHI-9i. 

## 4. Results 

### 4.1. Test–Retest Reliability 

After eliminating all incomplete questionnaires, 416 test–retest pairs were left. The mean age (±SD) was 50 (±17), with males skewing generally older at 56 (±16) compared to female patients at 46 (±17) years of age. A total of 26 participants (6.3%) were classified as elite vocal performers, 59 as professional voice users (14.2%), 78 as non-vocal professionals (18.7%) and 253 as non-vocal non-professionals (60.8%). An overview of the test–retest population is given in Figure 1 and Table 2. 

The median gap between test and retest was 2 days, with a mean of 3.3 days. The overall mean difference between VHI-9i scores (± SD) was very small at 0.25 (±3.52). Gender, voice use or age showed similarly minor differences (see Figure 2 and Table 2). 

A paired-sample *t*-test between the VHI-9i total scores showed no significant differences (*p* = 0.146). Test and retest scores also correlated very well (r = 0.919, *p* < 0.01), indicating a highly reliable questionnaire. Only 5% of the population had a difference larger than 7 points. Gender had no impact on the reliability of the questionnaire. The independent sample *t*-test for the absolute VHI-9i score difference between males and females was not significant (*p* = 0.589). The level of voice use did also not affect reliability. The Kruskal–Wallis H-test showed no significance between the four voice use classifications (*p* = 0.701). The absolute score differences lightly depended on age. For every year of life, the difference rose by 0.016 points (*p* = 0.028). 

### 4.2. Validation 

Of the 3661 participants remaining for VHI-9i validation, 1456 were male (39.8%) and 2205 were female (60.2%). The mean age (±SD) was 48 (±17), with males being on average slightly older at 50 (±18) years compared to females at 47 (±17) years of age. Vocal impairment was caused by functional dysphonia in 40.8% of the study population. Patients with organic dysphonia (50.8%) showed various pathologies: mostly lesions of the lamina propria (e.g., vocal fold nodules, polyps, cysts, Reinke’s edema), followed by benign and malignant changes of the epithelium (e.g., leukoplakia, papillomatosis, carcinoma), as well as neurogenic voice disorders (e.g., unilateral paralyses of the recurrent laryngeal nerve, spasmodic dysphonia). The remaining 8.4% were healthy subjects without dysphonia, mainly college applicants who presented to receive a vocal fitness examination, or prior to starting a profession associated with high vocal demands (e.g., teachers, singers, lecturers). The population pyramid and pathology classification are shown in Figure 3. 

As the test–retest examinations demonstrated, the reliability of VHI-9i scores is not affected by gender or voice use. Although statistically significant, the age dependency is so small that it can be neglected in clinical practice. Therefore, all further observations and calculations were conducted for the entire population of 3661 participants. Using the old VHI-9i classification scale based on percentiles [25], 15.5% of our participants had healthy voices (total score 0 ≤ 5), 25.7% mild dysphonia (6 ≤ 13), 32.3% moderate (14 ≤ 22) and 26.5% severe dysphonia (23 ≤ 36). Applying the same method to the current database, 25% of patients had a score between 0 and 9, 50% up to 16, and 75% up to 22 points. The severity distribution for the other vocal parameters can be found in Table 3. Regarding VHIs, 63 cases had to be excluded (*n* = 3598 instead of 3661), because these test subjects had marked this question outside or in-between the provided options for the severity levels, rendering them invalid. 

The size and mean of each severity category as well as the distribution of scores were notably different between parameters. The VHI-9i histogram shows a centered flat curve (skewness 0.063, kurtosis −0.90), the DSI is still centered but steeper (skewness −0.04, kurtosis 0.48) and the VEM is even steeper and skewed towards lower VEM values (skewness −1.08, kurtosis 1.94), with most patients falling into severity category 3 (Figure 4). 

The VHI-9i total scores correlated the most with the VHIs, even though ϱ was only moderate (ϱ = 0.592; see Table 4). All other parameters correlated notably weaker with the VHI-9i. The objective DSI and VEM were also moderately correlated to each other at ϱ = 0.663. The distribution of subjects into G and R severity levels was rather similar, while B showed a different result with over 50% of all cases falling into the “healthy” category. G and R also had the strongest correlation among each other (ϱ = 0.871), reinforcing clinical experience that G serves as the gold standard for hoarseness evaluations via the GRB scale. 

Figure 5 shows the distribution of VHI-9i total scores using the classifications for VHIs, DSI, VEM and G. The boxplots reveal a clear tendency: the higher the severity level, the higher the associated median. However, there is also a lot of overlap between the quartiles of different severity levels. This especially applies to DSI and VEM, which makes these parameters less suitable for VHI-9i classification. 

The ROC plots (Figure 6) also favor the VHIs as the best classifying index. DSI, VEM and G are visibly less suitable classifiers, because their curves are closer to the hypothetical diagonal through the ROC plot, signifying weaker discriminating performance.

The AUC results (Table 5) mirror the correlations of vocal parameters (compare Table 4). The best performance was achieved by the VHIs with excellent AUCs, followed by acceptable values for G. The parameters DSI and VEM turned out to be poor discriminators, with AUCs below 0.7. 

As shown by our reliability analysis, severity categories must be at least 7 points in size to account for significant changes and minimize the possibility of retest artifacts. Neither optimizing for sensitivity and specificity (Max J) nor correctly classified cases (Max CCC) alone produced classes that were all wide enough (>7 points). Apart from the VHIs, Max CCC even produced cut-off recommendations that would eliminate the lowest (VEM) or lowest and highest (DSI, G) severity categories (highlighted in Table 5). Since both methods did not produce plausible cut-off values or categories of reasonable size, medians between the Max J and Max CCC measurements had to be calculated. 

However, both median calculations did not always return the exact same result, which is why the J–CCC–Median cut-off values are expressed as ranges in Table 5. In general, the difference between both medians was below 0.25 points most of the time and very rarely exceeded 0.5 points. The medians for all vocal parameters agreed on the first boundary (i.e., between severity levels 0 and 1) at 7 or 8. Between “mild” and “moderate” (severity levels 1 and 2), the median recommendations ranged from 14 to 20. Except for the VEM, the medians led to a cut-off point between 26 and 28 for the boundary distinguishing “moderate” from “severe” impairment (i.e., severity levels 2 and 3).

## 5. Discussion

The VHI-9i short scale has proven to be a valuable diagnostic tool in our clinical practice for well over a decade. The total number of 17,660 consecutively completed questionnaires documented in our database confirms its high acceptance among patients and medical staff. In our test–retest analysis, the VHI-9i questionnaire demonstrated very high reliability independent of gender or voice use. Age had a minor influence, which we do not consider clinically relevant: For every year of life, the absolute score difference between test and retest increased by 0.016. If we applied that difference to the entire age range of our study population, the VHI-9i total score of an adolescent compared to a senior person would differ by about 1. The reliability analysis also showed that the severity classes for the VHI-9i need to be at least 7 points in size (2*SD of paired sample *t*-test), since only differences of 7 points and above account for significant changes and minimize the possibility of retest artifacts. Our interpretation of the ROC analysis had to consider this requirement. Unfortunately, neither optimizing for Max J nor Max CCC resulted in categories that were all large enough. Calculating the median between them for each cut-off point, however, yielded satisfactory results for clinical use. 

All classification ranges are listed in Table 6. The Median J method strikes a good balance between sensitivity, specificity and the minimum class width of 7 points. The new boundary of a score of 7 corresponds directly with the VHIs Median J result for healthy voices (class 0). Finding a reasonable upper boundary for severity level 1 is more difficult: using VHIs Median J (a score of 14) would result in a category that is too small. The median for the expert auditory–perceptual assessment (G) points towards an even higher boundary (a score of 19). Since we were trying to find a mid-point for our severity classes, we decided to use the upper boundary of the 50% quartile (a score of 16). The upper boundary for severity level 2 (moderate impairment) can be taken once again from the VHIs Median J row, placing class 2 between 17 ≤ 26 and class 3 between 27 ≤ 36. 

Compared to the old VHI-9i classification scale based on percentiles [25], the revised severity ranges classify more patients towards the lower categories. Severity level 3 is reduced by 4 points and is no longer the largest category. Level 1 and 2 start at higher class boundaries due to the size increase in level 0. 

The best correlation was observed between VHI-9i and VHIs, making the overall self-assessed vocal impairment the best candidate for the validation process. However, the VHI-9i did not correlate well with the two objective parameters DSI and VEM, and had only slightly higher correlations with GRB. This supports recent studies that all these vocal parameters measure different aspects of a patient’s voice and are neither mutually interchangeable nor redundant [31,36,41,43]. Due to the weak correlations, poor discriminating performance and sometimes impossible cut-off points, DSI, VEM and G ultimately had no part in our recommendation for the revised VHI-9i cut-off points. It is important to remember that the VHI-9i does not measure objective voice impairment (DSI) or vocal capacity (VEM), but personal suffering due to a subjectively perceived vocal handicap. None of the parameters allow conclusions to be drawn about the diagnoses or underlying causes of the voice disorder. 

### Study Limitations

Over 60% of our test–retest population were categorized as non-vocal non-professionals. Ideally, the study would have included more subjects with professional backgrounds in singing, acting or teaching, especially since establishing independence from voice use was one of our goals during the rest-retest analysis. A bigger population of elite vocal performers and professional voice users would have been preferrable, but does not represent the actual proportions of our clinic clientele. 

Furthermore, males are underrepresented in our study, so there may be participation bias. Despite the limited number of male subjects, we concluded that the VHI-9i was independent of gender, but a more balanced gender involvement would have been more representative. However, our clinical experience shows that women are generally more likely to see a doctor for voice problems. 

In addition, signal-to-noise ratio (SNR) analysis and signal typing are considered to be important for valid and reliable perturbation measurements [44,45,46]. Unfortunately, this functionality is not included in the DiVAS software, which was specified in our study design as the main tool for objective voice analysis. One of the fundamental limitations of the DSI is the inclusion of jitter without sufficient evaluation of the signal type. In general, only type 1 and 2 are considered viable for perturbation analysis. The 5% jitter cut-off applied in our study was established to exclude type 4 signals only [46]. However, the categorization of a small test sample (*n* = 40) revealed signal type 1 and 2 exclusively, even for patients with low DSI and high jitter values. Furthermore, the majority of SNR results were between 42 and 50 dB (“recommended”), with a smaller number between 30 and 42 dB (“acceptable”) [45]. Therefore, we believe that our exclusion criteria were sufficient to eliminate voices which are not suitable for perturbation analysis. We recognize that this estimate cannot be taken as proof for the entire dataset and plan to include SNR and signal typing analyses in our future studies from the outset. It should also be noted that jitter was only used for DSI calculation, which proved to be irrelevant for the main goal of our study, i.e., a revised VHI-9i classification. Therefore, our recommendations regarding VHI-9i severity categories should not have been distorted. 

Moreover, our initial ROC analysis produced boundary recommendations that were not feasible for diagnostic purposes. The resulting severity categories would have been either too small (<7 points) or would even not exist at all. Calculating the median between Max J and Max CCC is not a commonly used method for solving these problems. However, based on the frequent use of the VHI-9i in clinical investigations [18,19,20,21,22,31,36,41], it appears that the new classification will be a practical option for clinical settings. 

In general, the auditory-perceptual assessment of voices via GRB was conducted only by two experienced examiners. Safer larger group judgments were not made. Due to the enormous number of cases (*n* = 17,660) and over a decade of diagnostic voice recordings, a retrospective blinded voice evaluation with 4-5 raters was not an option. 

## 6. Conclusions

The VHI-9i is a reliable questionnaire which is independent of gender and professional voice use. Its dependency on age is negligible. Based on many years of clinical experience, it also has high acceptance among patients and medical staff, making it a valuable diagnostic tool.

The old cut-off values for the VHI-9i severity categories based on percentiles had to be adjusted. We recommend setting class 0 (healthy) between 0 ≤ 7, class 1 (mild impairment) between 8 ≤ 16, class 2 (moderate impairment) between 17 ≤ 26 and class 3 (severe impairment) between 27 ≤ 36. 

The subjective VHI-9i does not correlate well with objective vocal parameters (DSI, VEM) or subjective auditory–perceptual assessment (GRB), reinforcing the notion that all these parameters measure different dimensions of a patient’s voice and are neither mutually interchangeable nor redundant. 

## Figures and Tables

**Figure 1 jcm-10-03325-f001:**
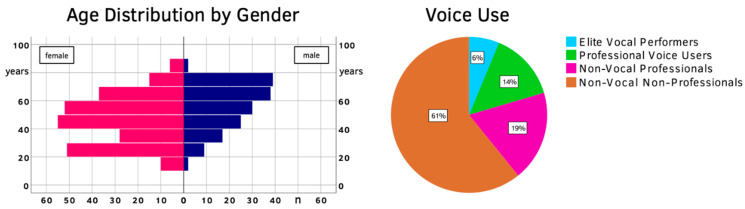
Overview of the test–retest population (age, gender, voice use classification).

**Figure 2 jcm-10-03325-f002:**
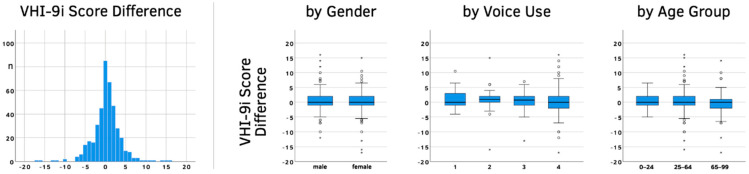
VHI-9i score difference between test and retest (total differences, by gender, by voice use, by age group). Age dependency was analyzed using discrete age values; age groups were only used in the diagram to improve the graphical representation. Circles (○) mark outliers (3rd quartile + 1.5*interquartile range; 1st quartile − 1.5*interquartile range) and asterisks (*) mark far outliers (3rd quartile + 3*interquartile range; 1st quartile − 3*interquartile range).

**Figure 3 jcm-10-03325-f003:**
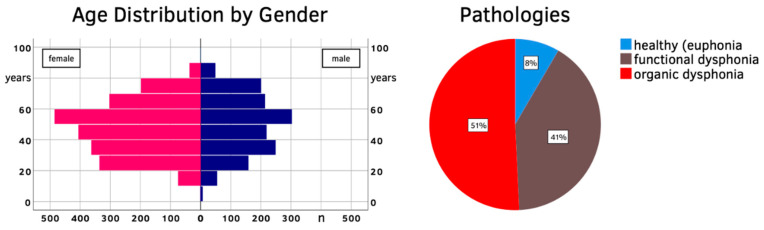
Overview of the validation population (age, gender, pathology classification).

**Figure 4 jcm-10-03325-f004:**
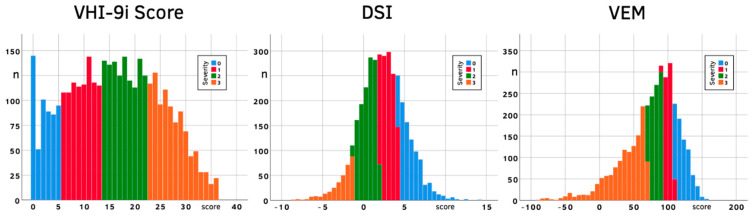
Observed VHI-9i, DSI and VEM scores with their associated severities.

**Figure 5 jcm-10-03325-f005:**
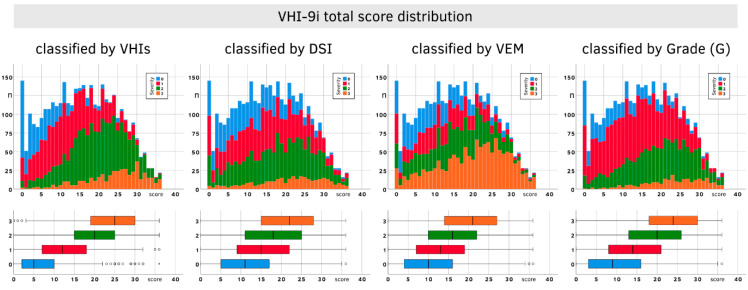
Distribution of VHI-9i total scores classified by VHIs, DSI, VEM and G severity levels. Upper row: stacked bar chart showing the number of subjects with their VHI-9i scores. Lower row: boxplots showing the percentiles of patients’ VHI-9i scores by severity level. Circles (○) and asterisks (*) mark outliers and far outliers.

**Figure 6 jcm-10-03325-f006:**
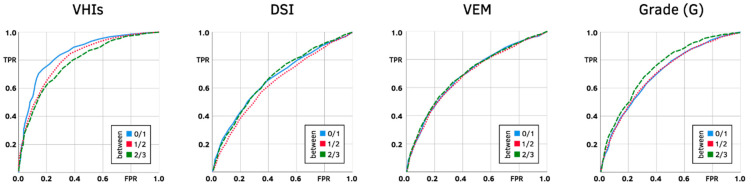
Combined ROC plots to determine cut-off points between severity categories 0 and 1 (blue), 1 and 2 (red), 2 and 3 (green).

**Table 1 jcm-10-03325-t001:** Severity classification of different vocal parameters, assessed by study participants (VHI-30, VHI-9i, VHIs), experienced clinicians (GRB), and acoustic–aerodynamic analysis (VEM, DSI). Although all parameters share the same classification scale (0–3), equal levels of severity among different parameters do not imply equivalence (***** classification ranges based on percentiles).

Level of Severity	VHI-30 *[25]	VHI-9i *[25]	VHIs	Grade (G)	VEM *[36]	DSI [38]
0: healthy	0 ≤ 14	0 ≤ 5	0	0	≥108	≥4.2
1: mild	15 ≤ 28	6 ≤ 13	1	1	93 < 108	1.8 < 4.2
2: moderate	29 ≤ 50	14 ≤ 22	2	2	69 < 93	−1.2 < 1.8
3: severe	51 ≤ 120	23 ≤ 36	3	3	<69	<−1.2

**Table 2 jcm-10-03325-t002:** Study participant distribution and VHI-9i score differences between test and retest.

	Number *n* (%)	Mean Total Score Difference (±SD)
Male	162 (38.9%)	0.38 (±3.68)
Female	254 (61.1%)	0.17 (±3.42)
Voice Use Level 1	26 (6.3%)	0.75 (±3.45)
Voice Use Level 2	59 (14.2%)	0.82 (±3.48)
Voice Use Level 3	78 (18.7%)	0.40 (±2.91)
Voice Use Level 4	253 (60.8%)	0.02 (±3.70)
Age Group 0–24 years	46 (11.1%)	0.41 (±2.17)
Age Group 25–64 years	267 (64.2%)	0.45 (±3.47)
Age Group 65–99 years	103 (24.7%)	−0.33 (±4.06)

**Table 3 jcm-10-03325-t003:** Collected voice data by vocal parameter, classified according to the associated level of severity as shown in Table 1.

Vocal Parameter		Level of Severity			
		0: Healthy	1: Mild	2: Moderate	3: Severe
VHIs	number (%) mean VHI-9i score (±SD)	559 (15.5%) 6.6 (±6.8)	1170 (32.5%) 12.8 (±7.2)	1425 (39.6%) 19.5 (±7.4)	444 (12.4%) 23.9 (±7.8)
DSI	number (%) mean VHI-9i score (±SD)	879 (24.0%) 11.9 (±8.1)	1210 (33.0%) 15.5 (±8.8)	1244 (34.0%) 17.7 (±8.9)	328 (9.0%) 21.0 (±8.5)
VEM	number (%) mean VHI-9i score (±SD)	732 (20.0%) 11.1 (±8.0)	673 (18.4%) 13.5 (±8.2)	945 (25.8%) 15.7 (±8.3)	1311 (35.8%) 19.9 (±8.8)
G	number (%) mean VHI-9i score (±SD)	537 (14.7%) 10.4 (±8.3)	1693 (46.2%) 14.2 (±8.4)	1169 (31.9%) 19.1 (±8.4)	262 (7.2%) 23.3 (±7.8)
R	number (%) mean VHI-9i score (±SD)	602 (16.4%) 11.7 (±8.9)	1864 (50.9%) 15.0 (±8.7)	1031 (28.2%) 18.9 (±8.4)	164 (4.5%) 21.8 (±8.2)
B	number (%) mean VHI-9i score (±SD)	1865 (50.9%) 12.8 (±8.4)	1205 (32.9%) 17.3 (±8.4)	446 (12.2%) 21.8 (±8.1)	145 (4.0%) 25.6 (±6.7)

**Table 4 jcm-10-03325-t004:** Results of correlation analysis between vocal parameters (Spearman’s rho). All correlation coefficients were significant (*p* < 0.001).

	VHIs (0–3)	DSI (0–3)	VEM (0–3)	G (0–3)	R (0–3)	B (0–3)
VHI-9i	0.592	0.292	0.373	0.393	0.299	0.386
VHIs (0–3)		0.229	0.261	0.328	0.263	0.287
DSI (0–3)			0.663	0.525	0.454	0.494
VEM (0–3)				0.494	0.390	0.501
G (0–3)					0.871	0.665
R (0–3)						0.449

**Table 5 jcm-10-03325-t005:** ROC results for potential cut-offs between severity categories (0–1, 1–2, 2–3) using Max J, Max CCC and Median calculations. Yellow cells mark impossible cut-offs. Median calculations for every ROC parameter (TPR, FPR, J, CCC) resulted in slightly different class boundaries, which were specified by the ranges of cut-off values.

		**VHIs**			**G**		
		**Cut 0–1**	**Cut 1–2**	**Cut 2–3**	**Cut 0–1**	**Cut 1–2**	**Cut 2–3**
AUC		0.846	0.811	0.783	0.704	0.709	0.748
Max J	TPR	0.737	0.781	0.743	0.633	0.664	0.683
	FPR	0.174	0.298	0.316	0.33	0.352	0.311
	J	0.564	0.483	0.427	0.303	0.311	0.372
	CCC	2702	2674	2486	2336	2394	2521
	cut-off	11.5	14.75	19.5	13.5	16.75	20.5
Max CCC	TPR	0.966	0.818	0.115	1	0.464	0
	FPR	0.651	0.337	0.014	1	0.193	0
	J	0.315	0.481	0.101	0	0.271	0
	CCC	3132	2675	3162	3124	2464	3399
	cut-off	2.5	13.5	32.5	0	21.25	36
J–CCC–Median	TPR	0.86	0.78	0.43	0.83	0.59	0.32
	FPR	0.35	0.3	0.1	0.56	0.28	0.09
	J	0.51	0.48	0.33	0.27	0.31	0.23
	CCC	2988	2674	3026	2813	2443	3182
	cut-off	7–8	14–15	26–27	7–8	19	28
		**DSI**			**VEM**		
		**Cut 0–1**	**Cut 1–2**	**Cut 2–3**	**Cut 0–1**	**Cut 1–2**	**Cut 2–3**
AUC		0.667	0.64	0.674	0.692	0.689	0.699
Max J	TPR	0.651	0.569	0.683	0.648	0.585	0.639
	FPR	0.39	0.344	0.416	0.35	0.296	0.329
	J	0.26	0.226	0.267	0.298	0.289	0.31
	CCC	2346	2266	2170	2373	2309	2415
	cut-off	13.5	17.25	17.75	13.5	16.75	17.75
Max CCC	TPR	1	0.408	0	1	0.786	0.44
	FPR	1	0.216	0	1	0.537	0.167
	J	0	0.193	0	0	0.25	0.273
	CCC	2782	2280	3333	2929	2425	2535
	cut-off	0	21.75	36	0	10.75	22.5
J–CCC–Median	TPR	0.83	0.48	0.3	0.83	0.66	0.53
	FPR	0.66	0.28	0.13	0.61	0.38	0.23
	J	0.17	0.2	0.17	0.22	0.28	0.3
	CCC	2604	2266	3012	2718	2360	2512
	cut-off	7–8	18–20	26–27	7–8	14	20–21

**Table 6 jcm-10-03325-t006:** Sizes of severity classes based on Max J, Max CCC and Median calculations. Green cells serve as the basis for our proposed new VHI-9i severity classification.

Classifying Method	Level of Severity			
	0: Healthy	1: Mild	2: Moderate	3: Severe
VHIs (Max J)	0 ≤ 12	13 ≤ 15	16 ≤ 20	21 ≤ 36
VHIs (Max CCC)	0 ≤ 3	4 ≤ 14	15 ≤ 33	34 ≤ 36
VHIs (Median J)	0 ≤ 7	8 ≤ 14	15 ≤ 26	27 ≤ 36
VHIs (Median CCC)	0 ≤ 8	9 ≤ 15	16 ≤ 27	28 ≤ 36
G (Max J)	0 ≤ 14	15 ≤ 17	18 ≤ 21	22 ≤ 36
G (Max CCC)	-	0 ≤ 21	22 ≤ 36	-
G (Median J)	0 ≤ 7	8 ≤ 19	20 ≤ 28	29 ≤ 36
G (Median CCC)	0 ≤ 8	9 ≤ 19	20 ≤ 28	29 ≤ 36
DSI (Max J)	0 ≤ 14	15 ≤ 17	18	19 ≤ 36
DSI (Max CCC)	-	0 ≤ 22	23 ≤ 36	-
DSI (Median J)	0 ≤ 8	9 ≤ 20	21 ≤ 27	28 ≤ 36
DSI (Median CCC)	0 ≤ 8	9 ≤ 18	19 ≤ 26	27 ≤ 36
VEM (Max J)	0 ≤ 14	15 ≤ 17	18	19 ≤ 36
VEM (Max CCC)	-	0 ≤ 11	12 ≤ 23	24 ≤ 36
VEM (Median J)	0 ≤ 8	9 ≤ 14	15 ≤ 20	21 ≤ 36
VEM (Median CCC)	0 ≤ 7	8 ≤ 14	15 ≤ 21	22 ≤ 36
VHI-9i quartiles	0 ≤ 9	10 ≤ 16	17 ≤ 22	23 ≤ 36
**Proposed new classification**	**0 ≤ 7**	**8 ≤ 16**	**17 ≤ 26**	**27 ≤ 36**

## Data Availability

All data of the study are available in the Department of Audiology and Phoniatrics, Charité-Universitätsmedizin Berlin, Berlin, Germany.

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
