# Peer review of "Validation and Classification of the 9-Item Voice Handicap Index (VHI-9i)"

_jcm, 2021, doi:10.3390/jcm10153325_

Round 1
Reviewer 1 Report
Thankyou for allowing me to review the manuscript “Validation and classification of the 9-item Voice Handicap Index (VHI-9i)”. It presents a large sample reliability and validation study of the VHI-9i using a range of acoustic measures, specifically DSI the VEM, and a self rating severity scale, the VHIs.
The paper is based on research conducted on a large clinical cohort. In my opinion, there are some significant research design issues, which I suspect are the reason for the null findings. There is an apparent bias towards the utility of the VHI-9i that comes through in the written expression and interpretation of the findings.
The introduction is well written but it fails to discuss the item and scale development of the VHI-9i in sufficient detail. The rationale for searching for cut-off score of severity is not explained. No other equivalent tool has been investigated in this way, I suspect due to the lack of criterion reference for severity. This would need to be established and might need to be the next step in addressing this research question.
The method as published in not replicable and not of sufficient scientific rigour. 30cm microphone is mouth distance is not recommended for voice source analysis. Please provide a reference for why this was selected. If jitter (or any F0 dependent perturbation measurement) is used in voice analysis, signal typing must be applied to ensure valid and reliable measurement. The diagnostic process for each of the participants is not described. Please add this for replicability.
What does "exceptionally great voices” mean – please use appropriately technical terminology.
Results : There is no need to provide Table and Figures for test-retest data - provide 1 or the other.
"A paired-samples t-test between the VHI-9i total scores showed no bias (p=0.146)" I don't understand this statement - paired T-tests don't analyse bias, they analyse if there is a significant difference between 2 means.
"Absolute score differences lightly depended on age. For every year of life, the difference rose by 0.016 points (p=0.028)." - this is interesting. Does it suggest we become more unreliable as we age?
“As the test-retest examinations demonstrated, the reliability of VHI-9i scores is not affected by gender, age or voice use in a relevant way.” The use of the term ‘relevant’ needs to be clarified as it seems age does based on the p value quoted previously. What determine if it is relevant or not.
“Regarding VHIs, 63 cases had to be excluded, because the rated impairment was in-between the given severity levels” - this needs to be explained. How is this possible?
Discussion :This is brief and mentions appropriate limitations. The conclusion is optimistic in interpretation of the results, which other than rest-retest reliability, are predominantly null findings. The authors sound somewhat biased toward the use of the VHI-9i.
Overall, I am somewhat perplexed by the decision to use the VEM, DSI, and GRB as referents to establish validity and discriminate severity cut-offs. VEM and DSI are of objective signs related to (mostly) voice signal features. GRB is a perceptual scale related to the voice signal. The VHI-9i measures symptoms reported by patients of disorder. It is no surprise that these are not good discriminators as they measure different constructs. The use of the VHIs seems more likely to correlate because it is a self reported measure so this would be expected to correlate well. However, the use of the VHIs as the main classification criterion is not advisable as this measure has not been tested for validity and reliability as a measure of severity. As the goal of this research seemed to be to establish cut-off scores for differing levels of severity, and the VHIs has not been proven to do this, it seems illogical to choose it as the preferred classification criterion. Also, the choice of using quartiles as divisions seems an arbitrary exploration as no rationale for this approach to clinical data is provided.
In my opinion, the above issues preclude the manuscript from being published.
As a side note, there is significant criticism in the literature of translations and adaptations of voice patient reported outcome measures (see Gilbert, Gartner-Schmidt, & Rosen (2017). I would suggest that the authors consider reviewing their data to assess the construct validity of the VHI-9i and VHIs to differentiate normal from disordered in the first instance and/or use an equivalent PROM (e.g. the VOQL) appropriately translated and tested as a criterion reference.
Author Response
Dear Reviewer 1,
thank you for your detailed and critical review. Please find our responses in the attached pdf.
Thank you for your time and helpful comments to improve the quality of the paper.
We hope that your main concerns have been addressed so that you agree with the publication of our study.
On behalf of all authors,
Philipp P. Caffier

Reviewer 2 Report
Overall, I think the manuscript is well written. I have several questions regarding the manuscript, though.
1) I just want to confirm "i.e." in the first phrase of Section 2.2 is as intended; if the versions described in the parentheses are all and there is no other version, it is just OK.
2) At the end of Section 4.1, I guess the dependency of test-retest difference on age was analyzed using regression. I think the authors should declare how they treated the factor of age, whether they used actual numerical value of age or they used three-level classes as described in Table 2.
3) In Figure 3, the positions of the axis title feel strange to me. I guess the authors can place them outside of the labels.
4) In the 6th line of Chapter 5, the authors pointed out the dependency of the absolute score difference between test and retest on age in the first half of the phrase, while they explained it as the difference of the total score between the adolescents and seniors in the last half. I think the dependency on age is not exactly same regarding the test-retest difference and the total score.
5) I am confused by the cut-off point derived from the median; in Chapter 4, I regard the cut-off point derived from median requires both cut-off by the best J and that by the best CCC, while there are two independent cut-off from each of median J and median CCC in Chapter 6. I am not sure whether these cut-offs derived from median are same or not. If these cut-offs were derived by different procedures, please describe them explicitly.
Author Response
Dear Reviewer 2,
thank you for your review. Please find our responses in the attached pdf.
Thank you for your time and helpful comments to improve the quality of the paper.
We hope that your main concerns have been addressed so that you agree with the publication of our study.
On behalf of all authors,
Philipp P. Caffier

Reviewer 3 Report
This is a well-written and well-conducted study that provides validation and classification of dysphonia severity for the VHI-9i. The sample size is huge, which allowed the authors to examine the data in multiple subgroups and with multiple instruments.
I have a few minor issues and questions for the authors:
- I am not familiar with the use of only GRB from the GRBAS scale; the A is often omitted, but dropping Strain from the assessment is surprising and atypical. Please provide an explanation. (If hoarseness is the only perceptual characteristic of interest, wherein hoarseness is defined as roughness + breathiness, then I suppose omitting strain is reasonable.)
- Grade is described as the “gold standard for hoarseness evaluations”, but I think a better phrase would be “overall indicator of dysphonia” and clarify that this is determined by expert listeners. Hoarseness, as stated, involves R+B, but G includes all aspects of dysphonia, including strain, asthenia, pitch and loudness anomalies. If G is used as the main indicator of severity of hoarseness, then it is especially confusing to read about “levels of severity” in the results – I had to read the methods several times to realize that severity was sometimes determined by the listeners and sometimes by the talkers. Please include information in each section of the paper to help readers understand this important distinction.
- It is interesting that the “Objective examination instruments” were all based on acoustic assessments (consider new heading name) and all but one (jitter) were based on maximum performance tasks. These tasks (MPT, min and max F0, min and max SPL) are not akin to typical speech tasks, yet the reading sample collected for the GRB ratings could have been subjected to acoustic analysis, particularly using cepstral/spectral techniques. For example, the cepstral-spectral index of severity (CSID) should correlate well with ratings of G. Given these task differences, it is not so surprising that the VHI-9i didn’t correlate well with the DSI or VEM.
- I found it confusing to see Level of Severity as column headings but Grade as a row in Table 3; at first, I thought these were the same thing. It would be helpful to readers to use “Levels of Self-Rated Severity” as the merged column heading. This is especially important when explaining that G does not correlate strongly with VHIs. (This relates to the comment in #2 above.)
- Tables and figures are often redundant. Consider keeping one and not the other.
- It was surprising to read the phrase “we are sure that the new classification will pass the practical test” under Study Limitations. This conveys a sense of “our results don’t really matter in the end.” Please reconsider this paragraph’s concluding sentence so as not to undermine the results of this very nice study.
- The body of the paper is written better than the abstract. The authors may want to review the abstract carefully for writing style, flow of information, and clarity, since it is arguably the most important part of the article.
Thank you for the opportunity to review this very nice article.
Author Response
Dear Reviewer 3,
thank you for your review. Please find our responses in the attached pdf.
Thank you for your time and helpful comments to improve the quality of the paper.
We hope that your main concerns have been addressed so that you agree with the publication of our study.
On behalf of all authors,
Philipp P. Caffier

Round 2
Reviewer 1 Report
Thankyou for your response. You have addressed the majority of issues and made appropriate changes in the text. IMO 2 issues remain with the manuscript that need to be addressed.
The method still does not provide sufficient technical detail to replicate the study:
"For objective external validation criteria, we applied acoustic-aerodynamic voice function diagnostics. Voice recordings of all participants were conducted at the voice lab of our outpatient department, which is a sound-treated room with a background noise <40 dB(A). Study participants were asked to wear a head-mounted microphone with a stable mouth-microphone distance of 30 cm [32]"
Missing information includes: all equipment used (manufacturer & model number), Freq response of microphone, equipment settings. I do not think it is appropriate for me to read another 2 journal articles to glean this essential information.
The second issue regards the authors response to my concerns about the validity of the jitter measures. In your response you state that :
" as long as the signal-to-noise ratio is greater than about 40 dB and concomitant amplitude modulations are below about 5% " after which you then discuss something else. Nowhere in the data analysis do you assess SNR to provide evidence that your acoustic analysis was valid. Recent recommendations have stated that only samples with SNR ≥ 30dB should be used for acoustic analysis. (See Deliyski, D.D.; Shaw, H.S.; Evans, M.K. Adverse effects of environmental noise on acoustic voice quality measurements. J Voice 2005, 19, 15-28, )
The argument provided to ignore my recommendation for the need to signal type (Titze et al, 1993) is very old and has been superseded by himself (Titze, I. 1995. “Workshop on acoustic voice analysis: Summary statement,”National Center for Voice and Speech, Denver, CO.) and again by Sprecher et al, 2010.
One of the fundamental limitations of the DSI is that it calculates Jitter without sufficient evaluation of the signal to ensure valid measurement. Sprecher et al demonstrate that 5% jitter cut-off is only sufficient to exclude Type 4 signals. So combined with the issue with SNR cut off at 40dB, it is highly likely that the analysis of perturbation in this study is not reliable.
Given the importance of this analysis to the research, I would recommend the authors do the work of ensuring that their voice samples are appropriate for analysis by following recommendations.ie reanalyse the data for SNR (PRAAT can be used) and signal typing (Bridge2Practice can be used).
All the other concerns raised have been addressed sufficiently.
Thankyou for the opportunity to review this manuscript.
Author Response
Dear Sir or Madam,
attached please find the second revision of our manuscript entitled “Validation and classification of the 9-item Voice Handicap Index (VHI-9i)”, which we hope you will find now suitable for publication in the “Journal of Clinical Medicine”.
The new content of the second revision is presented in green color (red: old modifications of the first revision).
Sincerely,
Philipp Caffier
Reviewer 2 Report
I appreciate the authors' revised manuscript and the cover letter. I am satisfied with the revisions and the comments. I agree to accept the current revised manuscript.
Author Response
Dear Reviewer 2,
thank you for your response.
We are glad that we could address all of your concerns.
Sincerely,
on behalf of all authors,
Philipp Caffier
Reviewer 3 Report
Thank you for the thorough responses to my comments and for the tutorial about the GRBAS scale and it subsequent iterations. The revisions are appropriate and acceptable except for the use of the term aerodynamic (see comment regarding Section 2.3 below). Minor wording suggestions follow.
Abstract: Sentence 3: “The main goals of this study were to establish … levels.”
Section 2.2., paragraph 3, line 4: Note the typo: grad instead of grade
Section 2.3. I take issue with the implication that aerodynamic assessment was included in this study. MPT simply that – the duration of a maximum phonation task. It is not a measure of lung volume, lung pressure, or air flow; these are required in order for the measure to be aerodynamic. I realize that this term was included in the previous version of the paper, but it seemed primarily descriptive; now, it comes across as a primary measure. I would simply remove it. The DSI and VEM are both purely based on acoustic measures.
Study Limitations. The revised sentence about “the practical test” may have overshot the intended message. Try this instead: “However, based on the frequent use of the VHI-9i in clinical investigations [18-22, 31, 36, 41], it appears that the new classification will be a practical option for clinical settings.”
Author Response
Dear Reviewer 3,
thank you for your response.
We agree and implemented your suggestions (Abstract, Section 2.2., and Study Limitations.). As for Section 2.3., we deleted the word "aerodynamic" as requested. The new section header reads: "2.3. Objective Acoustic Assessment" in order to differentiate it from the previous heading ("2.2. Subjective Examination Instruments").
We hope that all your concerns have been addressed.
Sincerely,
on behalf of all authors,
Philipp Caffier